# Direct Inhibition of the Allergic Effector Response by Raw Cow’s Milk—An Extensive In Vitro Assessment

**DOI:** 10.3390/cells9051258

**Published:** 2020-05-19

**Authors:** Suzanne Abbring, Bart R. J. Blokhuis, Julie L. Miltenburg, Kiri G. J. Romano Olmedo, Johan Garssen, Frank A. Redegeld, Betty C. A. M. van Esch

**Affiliations:** 1Division of Pharmacology, Utrecht Institute for Pharmaceutical Sciences, Faculty of Science, Utrecht University, 3584 CG Utrecht, The Netherlands; s.abbring@uu.nl (S.A.); b.r.j.blokhuis@uu.nl (B.R.J.B.); juliemiltenburg@gmail.com (J.L.M.); k.g.j.romanoolmedo@students.uu.nl (K.G.J.R.O.); j.garssen@uu.nl (J.G.); f.a.m.redegeld@uu.nl (F.A.R.); 2Immunology Platform, Danone Nutricia Research, 3584 CT Utrecht, The Netherlands

**Keywords:** allergic diseases, FcεRI, mast cell, milk fractionation, milk processing, raw cow’s milk

## Abstract

The mechanisms underlying the allergy-protective effects of raw cow’s milk are poorly understood. The current focus is mainly on the modulation of T cell responses. In the present study, we investigated whether raw cow’s milk can also directly inhibit mast cells, the key effector cells in IgE-mediated allergic responses. Primary murine bone marrow-derived mast cells (BMMC) and peritoneal mast cells (PMC), were incubated with raw milk, heated raw milk, or shop milk, prior to IgE-mediated activation. The effects on mast cell activation and underlying signaling events were assessed. Raw milk was furthermore fractionated based on molecular size and obtained fractions were tested for their capacity to reduce IgE-mediated mast cell activation. Coincubation of BMMC and PMC with raw milk prior to activation reduced β-hexosaminidase release and IL-6 and IL-13 production, while heated raw milk or shop milk had no effect. The reduced mast cell activation coincided with a reduced intracellular calcium influx. In addition, SYK and ERK phosphorylation levels, both downstream signaling events of the FcεRI, were lower in raw milk-treated BMMC compared to control BMMC, although differences did not reach full significance. Raw milk-treated BMMC furthermore retained membrane-bound IgE expression after allergen stimulation. Raw milk fractionation showed that the heat-sensitive raw milk components responsible for the reduced mast cell activation are likely to have a molecular weight of > 37 kDa. The present study demonstrates that raw cow’s milk can also directly affect mast cell activation. These results extend the current knowledge on mechanisms via which raw cow’s milk prevents allergic diseases, which is crucial for the development of new, microbiologically safe, nutritional strategies to reduce allergic diseases.

## 1. Introduction

Mast cells are granular immune cells crucial in type I hypersensitivity reactions. IgE-mediated allergic reactions are induced when allergens cross-link allergen-specific IgE antibodies bound to high-affinity IgE receptors (FcεRI) expressed on the mast cell surface [1,2]. This cross-linking causes aggregation of the FcεRI, which triggers a cascade of intracellular signaling pathways resulting in mast cell activation [3]. Upon activation, mast cells secrete various soluble mediators generally divided in three categories; preformed mediators stored in the cell’s cytoplasmic granules (e.g., histamine, heparin, proteases), de novo generated lipid mediators (e.g., prostaglandins, leukotrienes, platelet-activating factor), and de novo synthesized cytokines and chemokines (e.g., TNFα, IL-6) [4]. By inducing vasodilation, vascular permeability, smooth muscle contraction, and mucus secretion, these mediators are responsible for the initiation and exacerbation of allergic symptoms [3].

Given the pivotal role of mast cells in allergic diseases, the inhibition of IgE-mediated mast cell activation is a common therapeutic strategy. Several pharmacological agents have been developed to block mast cell mediator receptors on target cells (e.g., antihistamines), to inhibit mediator synthesis (e.g., steroids, nonsteroidal anti-inflammatory drugs), and to prevent IgE-driven mast cell activation (e.g., Omalizumab) [2]. However, these drugs have varying success rates, making strict allergen avoidance still the only effective treatment [5].

Another approach is to reduce allergic sensitization and to favor immune tolerance, for example, by promoting aspects of lifestyle that seem to reduce the risk of allergic diseases based on epidemiological studies. The consumption of raw, unprocessed cow’s milk is such an aspect. Epidemiological studies have consistently shown an inverse association between raw cow’s milk consumption and the development of allergic diseases [6,7,8,9,10]. These findings were strengthened by causal evidence showing the potency of raw cow’s milk to reduce/prevent allergic diseases [11,12,13]. As source of immunomodulatory components, raw cow’s milk is speculated to exert its allergy-protective effects by creating a tolerogenic environment favoring unresponsiveness upon allergen exposure [14,15,16]. Recently, we provided evidence to support this hypothesis by demonstrating, in a murine model for food allergy, that the suppression of allergic symptoms by raw cow’s milk was accompanied by a reduction in allergen-specific Th2 cell responsiveness and an induction of tolerance-associated cell types, like CD103^+^ dendritic cells and regulatory T cells [12].

Currently, the allergy-protective effects of raw cow’s milk are therefore mainly attributed to the capacity of the milk to modulate T cell responses. However, the key role of mast cells in allergic diseases raises the question of whether raw cow’s milk can also directly influence the allergic effector response by targeting mast cell activation and function. To investigate this, the effects on in vitro IgE-mediated mast cell activation were studied. Since there is increasing evidence showing that the protective effects of raw cow’s milk are abolished upon milk processing [6,11,17], modulation by heated raw milk and shop milk (store-bought milk) was also assessed. In addition, raw milk was fractionated based on molecular size and effects of raw milk fractions on mast cell activation were evaluated to gain more insight into the components contributing to the protection against allergic diseases.

## 2. Materials and Methods

### 2.1. Isolation and Culture of Primary Mouse Mast Cells

Primary mast cells were generated from naïve female C_3_H/HeOuJ mice. To obtain bone marrow-derived mast cells (BMMC, Kleinmachnow, Germany), as representatives of mucosal-type mast cells, femurs and tibiae were removed and thoroughly flushed with culture medium (RPMI 1640 medium (Lonza, Verviers, Belgium) supplemented with 10% fetal bovine serum (FBS; Bodinco, Alkmaar, The Netherlands), 100 U/mL penicillin and 100 µg/mL streptomycin (Sigma-Aldrich, Zwijndrecht, The Netherlands), 20 mM Hepes, 0.1 mM MEM non-essential amino acids, 2 mM GlutaMAX, 1 mM sodium pyruvate and 50 µM 2-mercaptoethanol (all purchased from Gibco, Thermo Fisher Scientific, Paisley, Scotland, UK)), by using a needle and syringe. The suspension of bone marrow cells was passed through a 70 µm nylon cell strainer, centrifuged (380× *g*, 6 min), and incubated with hypotonic lysis buffer (8.3 g NH_4_Cl, 1 g KHC_3_O, and 37.2 mg EDTA dissolved in 1 L demi water, filter sterilized) to remove red blood cells. Cell suspensions were subsequently resuspended in freezing medium consisting of 40% culture medium, 10% dimethyl sulphoxide (DMSO; Sigma-Aldrich), and 50% FBS and stored in liquid nitrogen until culture. For each culture, bone marrow cells were thawed and cultured in culture medium supplemented with 20 ng/mL IL-3 and stem cell factor (SCF; ProSpec, Ness-Ziona, Israel) at 37 °C. Subsequently, half of the culture medium was refreshed weekly, supplemented with 10 ng/mL IL-3 and SCF. Cell density was kept at 1–1.5 × 10^6^ cells/mL. BMMC were used for experiments after 4 to 8 weeks of culturing. Peritoneal mast cells (PMC), as representatives of connective tissue-type mast cells, were generated as follows; the peritoneal cavity was washed with cold PBS and cells were collected. After lysing red blood cells, cells were cultured in culture medium supplemented with 50 ng/mL IL-3, IL-4, and SCF (ProSpec). After 1 week of culturing, nonadherent cells were collected and cultured in fresh culture medium [18]. Subsequently, half of the culture medium was refreshed twice a week. PMC were used for experiments after 4 weeks of culturing.

### 2.2. Milk Types

The raw cow’s milk used was collected from a biodynamic dairy farm (Hof Dannwisch, Horst, Germany). Upon collection, part of the raw milk was aliquoted and stored at −20 °C until further use. The remainder was heated in a water bath for 10 min at 80 °C, cooled to room temperature, aliquoted, and then stored at −20 °C until further use (heated raw milk). The shop milk used was a pasteurized and homogenized milk, standardized at 3.8% fat (EDEKA, Aachen, Germany). To keep treatments equal, this milk was also aliquoted and stored at −20 °C. The day before experiments were conducted, the different milk types were put overnight in the fridge to keep the thawing process constant. Just before use, they were placed in a water bath for 30 min at 37 °C to adjust the temperature to culturing conditions and to obtain homogeneous solutions.

### 2.3. Milk Fractions

To fractionate raw cow’s milk based on molecular size, qEV size exclusion columns (Izon Science, Oxford, UK) were used according to the manufacturer’s protocol. Briefly, 0.5 mL of 10,000× *g* raw milk supernatant (free of cells, cell debris, and cream) was loaded onto the size exclusion column (Izon Science) and the first 3 mL of eluent was discarded. Eluent fractions of 0.5 mL were then collected up to 12 mL (24 fractions), by continuously adding RPMI 1640 medium without l-glutamine and phenol red (Lonza) to the column. Protein content of each fraction was quantified by using a NanoDrop ND-1000 spectrophotometer (A280; Thermo Fisher Scientific). To determine the molecular weight of the proteins in each fraction, proteins were separated by using a 12.5% SDS-PAGE under non-reducing conditions and visualized with SYPRO^®^ Ruby Protein Gel Stain (Bio-Rad, Veenendaal, The Netherlands). Fractions were stored at −80 °C until further use.

### 2.4. Mast Cell Activation Assay

BMMC (1 × 10^6^ cells/mL) and PMC (3.2 × 10^5^ cells/mL) were incubated overnight with 5% *v*/*v* raw milk, heated raw milk, or shop milk at 37 °C. After washing 3 times with assay medium (RPMI 1640 medium without l-glutamine and phenol red (Lonza), supplemented with 1% FBS (Bodinco) and 2 mM GlutaMAX (Gibco, Thermo Fisher Scientific)), cells were primed with 10%–20% *v*/*v* 2,4-dinitrophenol (DNP)-specific IgE (culture supernatant of IgE producing hybridoma cells, clone 26.82), for 1 h at 37 °C. Subsequently, cells were washed twice and stimulated by a range of DNP-HSA (DNP conjugated to human serum albumin; Sigma-Aldrich) concentrations (BMMC: 0–100 ng/mL; PMC: 0–12.5 ng/mL), for 1 h at 37 °C. In addition, BMMC were also stimulated by a range of rat anti-mouse IgE mAb concentrations (BD Biosciences, Alphen aan de Rijn, The Netherlands; 0–125 ng/mL) and by ionomycin (1 µM; Sigma-Aldrich). The magnitude of mast cell activation was determined by measuring β-hexosaminidase (β-hex) and cytokine release. Β-hex release was quantified by measuring fluorescence (excitation 350 nm/emission 460 nm) with a Fluoroskan Ascent^®^ Microplate Fluorometer (Thermo Fisher Scientific), after incubating cell-free supernatant with 4-methylumbelliferyl N-acetyl-β-d-glucosaminide (4-MUG; 158 µM; Sigma-Aldrich) in citrate buffer (0.1 M, pH 4.5; Acros Organics, Geel, Belgium) for 1 h at 37 °C and terminating the enzymatic reaction by adding glycine buffer (0.1 M, pH 10.7; Merck, Darmstadt, Germany). Maximum β-hex release was determined by lysing the cells with 0.5% Triton X-100 (Sigma-Aldrich). The percentage of β-hex release was calculated using the following formula:$$\frac{\left(A-B\right)}{\left(T-B\right)}\times 100\%$$
where A is the amount of β-hex released from stimulated cells, B the amount released from unstimulated cells, and T the amount of β-hex released from Triton X-100 lysed cells. Cytokine production was determined in cell-free supernatant harvested 18 h after DNP-HSA stimulation. IL-6 concentrations were analyzed using a mouse IL-6 ELISA MAX Standard Set (Biolegend, San Diego, CA, USA), according to the manufacturer’s instructions. IL-13 concentrations were analyzed by means of ELISA, as described elsewhere [19]. For raw milk fractions, BMMC were incubated for 2 h with the fractions, supplemented with 10% bovine serum albumin (BSA; Sigma-Aldrich) and 1% GlutaMAX. Mast cell activation was performed as described above.

### 2.5. Flow Cytometric Analysis of BMMC

After overnight milk incubation, BMMC were washed and incubated with anti-mouse CD16/CD32 (Mouse BD Fc Block; BD Biosciences) for 15 min on ice, to block non-specific binding sites. Cells were subsequently stained with FcεRI-PE-Cy7 and CD117-APC antibodies (Thermo Fisher Scientific) for 45 min on ice. Viable cells were distinguished using the dead cell dye YO-PRO^®^-1 Iodide according to the manufacturer’s instructions (Thermo Fisher Scientific). Cell viability and expression of FcεRI and CD117 were measured on the FACS Canto II (BD Biosciences) and analyzed with FlowLogic Software (Inivai Technologies, Mentone, Australia). Isotype controls were used and cut-off gates for positivity were determined with fluorescence-minus-one controls.

### 2.6. Analysis of Calcium Flux

BMMC were incubated overnight with the different milk types, washed, and primed for 1 h with 10%–20% *v/v* DNP-specific IgE (culture supernatant of IgE-producing hybridoma cells, clone 26.82), as described above. Cells were then washed again and loaded with the calcium-sensitive dye Fluo-4, AM (4 µM; Invitrogen, Thermo Fisher Scientific), by incubating them at 37 °C for 30 min, followed by 30 min at room temperature. After Fluo-4, AM loading, cells were washed and incubated for 30 min at room temperature with RPMI 1640 medium (without l-glutamine and phenol red; Lonza, Verviers, Belgium)/1% FBS (Bodinco). Prior to stimulation with DNP-HSA, baseline fluorescent readings were measured in 4 s intervals for 1 min using a Fluoroskan Ascent^®^ Microplate Fluorometer, with 492 nm excitation and 518 nm emission filters (Thermo Fisher Scientific). Cells were then treated with DNP-HSA (12.5 ng/mL; Sigma-Aldrich) or RPMI 1640 medium/1% FBS (as a control) and fluorescence was measured in 10 s intervals for 7 min.

### 2.7. Immunoblotting for Membrane-Bound IgE Expression and SYK and ERK Phosphorylation

For the determination of membrane-bound IgE expression, BMMC were lysed for 15 min on ice with PBS/0.5% Triton X-100 (Sigma-Aldrich) buffer supplemented with protease inhibitors (cOmplete, Mini Protease Inhibitor Cocktail; Roche Diagnostics, Mannheim, Germany) after incubation with the different milk types and IgE-mediated activation, as described earlier. After centrifugation for 10 min at 4000× *g*, supernatant was collected, and SDS sample loading buffer (58.3 mM Tris-HCl (pH 6.8), 6% *v*/*v* glycerol, 1.7% *w*/*v* SDS, 0.01% *w*/*v* bromophenol blue, and 100 mM DTT) was added. For the determination of spleen tyrosine kinase (SYK) and extracellular signal-regulated kinase (ERK) phosphorylation, BMMC were incubated for 3 h at 37 °C with RMPI 1640 medium (without l-glutamine and phenol red; Lonza)/0.2% BSA (Sigma-Aldrich) after milk incubation and IgE priming, to deplete them from serum. Cells were subsequently washed in PIPES buffer (140 mM NaCl, 5 mM KCl, 1 mM MgCl, 5.6 mM glucose, 10 mM PIPES, and 1.4 mM CaCl_2_·2H_2_O) and stimulated for 10 min at 37 °C by DNP-HSA (0–100 ng/mL; Sigma-Aldrich), diluted in PIPES buffer. Phosphorylation was stopped and cells were lysed by adding SDS sample loading buffer supplemented with protease inhibitors (cOmplete, Mini Protease Inhibitor Cocktail; Roche Diagnostics), phosphatase inhibitors (phosSTOP; Roche Diagnostics), 5U benzonase nuclease (Merck), 1 mM MgCl_2_ and 0.5 mM AEBSF (Merck). Upon addition of loading buffer, samples (for both protocols) were boiled for 5 min and loaded onto a 4%–20% precast polyacrylamide gel (Bio-Rad). After running the gels, proteins were transferred to a polyvinylidene fluoride membrane (Bio-Rad) and non-specific binding was blocked by incubation with TBS containing 0.1% Tween 20 (TBS-T) and 5% non-fat dry milk for 1 h at room temperature. Membranes were then incubated with HRP-conjugated goat anti-mouse IgE antibody (SouthernBiotech, Birmingham, AL, USA) in TBS-T/5% milk for 1 h at room temperature, or with rabbit anti-mouse phospho-SYK (Tyr525/526 antibody (Cell Signaling Technology, Leiden, The Netherlands) or rabbit anti-mouse phospho-p44/42 MAPK antibody (ERK1/2; Cell Signaling Technology) in TBS-T/5% BSA overnight at 4 °C. The membrane detecting SYK and ERK phosphorylation was subsequently incubated with HRP-conjugated goat anti-rabbit antibody (DAKO, Eindhoven, The Netherlands) in TBS-T/5% milk for 1 h at room temperature. HRP-detected protein bands were visualized using Clarity Western ECL Substrate (Bio-Rad) and imaged and quantified by ChemiDoc MP Imaging System (Bio-Rad) and Image Lab Software (version 5.2; Bio-Rad). The membrane detecting SYK phosphorylation was reprobed with the primary antibody for pERK after stripping the membrane with Restore Western Blot Stripping Buffer (Thermo Fisher Scientific). The expression of IgE and phosphorylated SYK (pSYK) and ERK (pERK) was normalized using mouse β-actin (Cell Signaling Technology).

### 2.8. Statistical Analysis

Results are expressed as mean ± SEM and differences compared to the control group were statistically determined, using one-way ANOVA followed by Dunnett’s multiple comparisons test. For Western blot results, immunoblots and the corresponding densitometric values are displayed. To analyze the calcium flux, the area under the curve (AUC) was calculated. Results were considered statistically significant when *p* < 0.05. All statistical analyses were performed using GraphPad Prism software (version 7.03; GraphPad Software, San Diego, CA, USA).

## 3. Results

### 3.1. Raw Milk Inhibits IgE-Mediated Mast Cell Activation

To determine whether raw cow’s milk affects IgE-mediated mast cell activation, BMMC were sensitized with DNP-specific IgE and subsequently cross-linked with DNP-HSA to stimulate degranulation. As expected, DNP-HSA concentrations ranging from 0.78 to 50 ng/mL induced a dose-dependent increase in β-hex release (Figure 1A). Incubating BMMC overnight with raw milk prior to mast cell activation reduced the β-hex release by approximately 35% (*p* < 0.05; Figure 1A). Pre-treatment of BMMC with heated raw milk or shop milk had no effect (Figure 1A). Similar results were observed when anti-mouse IgE mAb was used to cross-link IgE bound to FcεRI to stimulate degranulation (Appendix A). BMMC are considered to represent mucosal-type mast cells. The other main murine mast cell phenotype, the connective tissue-type mast cells, are represented by PMC [20]. To assess whether raw milk exerts similar effects on both phenotypes, PMC were also activated in an allergen-specific manner. Because of their higher sensitivity, lower allergen concentrations were used (Figure 1B). Incubating PMC overnight with raw milk resulted in a comparable β-hex reduction of approximately 40% (*p* < 0.05; Figure 1B). Again, heated raw milk and shop milk showed no effect (Figure 1B). In addition, the effect of raw milk on de novo cytokine production upon mast cell activation (12.5 ng/mL DNP-HSA) was investigated. Consistent with the reduction in β-hex release, raw milk caused a 50% reduction in BMMC-produced IL-6 (*p* = 0.0118; Figure 1C). IL-13 concentrations were even reduced by 75% in raw milk-treated BMMC (*p* = 0.0424; Figure 1D). No inhibition in IL-6 and IL-13 production was observed after incubation with heated raw milk or shop milk. IgE-mediated stimulation of PMC did not result in detectable cytokine production (data not shown). In further experiments, BMMC were used to further delineate the mechanism of inhibition by raw milk.

### 3.2. Inhibition of IgE-Mediated Mast Cell Activation Is Not due to Lower Expression of FcεRI on Mast Cells or Decreased Viability

To confirm that the inhibition of IgE-mediated mast cell activation by raw milk was not due to lower receptor expression or reduced cell viability, FcεRI and CD117 expression and BMMC viability after overnight milk incubation were analyzed by flow cytometry. Untreated BMMC showed around 90% viability and approximately 90% of these cells expressed FcεRI and CD117 (Figure 2A,B). BMMC viability (Appendix A) and FcεRI and CD117 expression (Figure 2B) were not affected by any of the milk types.

### 3.3. Ionomycin-Induced Mast Cell Activation Is Not Affected by Raw Milk Exposure

In order to investigate whether raw milk incubation only reduced FcεRI-triggered mast cell activation, BMMC were activated with the calcium ionophore ionomycin. Ionomycin directly transports Ca^2+^ across the cell membrane and rapidly depletes intracellular calcium stores, thereby artificially increasing intracellular calcium levels and bypassing the proximal signaling pathway of the FcεRI [21]. In Figure 2C, it is shown that neither raw milk nor heated raw milk or shop milk significantly affected ionomycin-induced β-hex release, suggesting that raw milk may inhibit proximal steps of IgE-induced mast cell degranulation.

### 3.4. Raw Milk-Treated BMMC Retain Membrane-Bound IgE Expression after Allergen-Specific Stimulation

To further understand the inhibitory effect of raw milk on FcεRI-induced mast cell activation, responses on parts of the associated signaling cascade were assessed. Upon allergen cross-linking, FcεRI complexes translocate into glycolipid-enriched membrane domains, better known as lipid rafts. This translocation is followed by rapid FcεRI internalization resulting in the removal of the receptor from the cell membrane [22,23]. To investigate whether raw milk reduces IgE-mediated BMMC activation by preventing FcεRI migration to lipid rafts and subsequent internalization, membrane-bound IgE expression upon DNP-HSA stimulation was determined. As expected, control BMMC showed a reduction in the presence of membrane-bound IgE with increasing DNP-HSA concentrations (Figure 3A,B). Interestingly, raw milk-treated BMMC retained IgE expression on their cell membrane after activation, regardless of DNP-HSA concentration (Figure 3A,B). Heated raw milk and shop milk-exposed BMMC showed a similar reduction in membrane-bound IgE expression density as control BMMC (Figure 3A,B), which tended to be lower than the membrane-bound IgE expression on raw milk-treated BMMC (Figure 3C).

### 3.5. Decreased Calcium Influx upon Allergen Challenge in BMMC Exposed to Raw Milk

The elevation of cytoplasmic calcium levels is key in the activation pathway leading to mast cell degranulation [3]. To assess whether the inhibition of IgE-mediated BMMC activation by raw milk coincided with a reduced calcium influx, BMMC were loaded with the calcium-sensitive dye Fluo-4, and then exposed to DNP-HSA. Changes in Fluo-4 fluorescence, corresponding to changes in intracellular calcium levels, were subsequently examined. As shown in Figure 4A, control cells showed a rapid rise in intracellular calcium levels upon allergen challenge (12.5 ng/mL DNP-HSA). This increase peaked at about 60 s, gradually decreased thereafter, and tended to plateau after approximately 300 s (Figure 4A). No calcium influx was observed when control cells were not stimulated (Figure 4A). When BMMC were preincubated with raw milk, the calcium influx was inhibited, as shown by a significant reduction in the AUC (*p* = 0.0167; Figure 4B). BMMC preincubated with heated raw milk or shop milk showed similar intracellular calcium levels upon DNP-HSA challenge as control cells (Figure 4A,B).

### 3.6. Reduced Cytokine Production after Raw Milk Treatment Coincided with Lower ERK Phosphorylation

Whereas an increase in intracellular calcium levels is crucial for mast cell degranulation, activation of the mitogen-activated protein kinase (MAPK) pathway is of importance for de novo synthesis and the secretion of cytokines. The MAPK proteins ERK, p38, and c-Jun N-terminal kinase (JNK) regulate the phosphorylation of specific transcription factors important for the synthesis of cytokines [1]. To examine whether the observed reduction in IL-6 and IL-13 production by raw milk-treated BMMC was the result of lower MAPK activation, ERK phosphorylation upon DNP-HSA stimulation was determined. Western blotting showed a lower ERK1/2 phosphorylation in the raw milk group compared to the control group (Figure 5A–C), which is in accordance with the reduction in cytokine production by raw milk-treated BMMC. For heated raw milk and shop milk, ERK1/2 phosphorylation levels were comparable to levels observed in untreated BMMC, which also corresponds to cytokine results (Figure 5A–C). For SYK, a protein more proximal in the FcεRI-induced signaling cascade, comparable results were observed with lower phosphorylation levels in the raw milk group compared to the heated raw milk and shop milk group (Figure 5D–F).

### 3.7. Raw Milk Fractions 2 and 3 Capable of Reducing IgE-Mediated Mast Cell Activation

To gain more insight into the raw milk components responsible for the reduced mast cell activation, raw milk was fractionated based on molecular size and 24 raw milk fractions were tested for their capacity to reduce allergen-induced β-hex release. Figure 6A shows the chromatogram of proteins eluded from the gel-filtration column, with a protein peak at fractions 2–5 and around fraction 18. Raw milk fractions were subsequently fractionated by SDS-PAGE and proteins were visualized using SYPRO^®^ Ruby Protein Gel Stain, to determine the molecular weight of the proteins in each fraction. Fractions 2–5 contained mostly proteins with a molecular weight between 50 and 75 kDa (Figure 6B). From fraction 11 onwards, smaller proteins with a molecular weight of around 10 to 25 kDa were more abundantly present (Figure 6B). When tested for their effect on IgE-mediated mast cell activation, raw milk fraction 2 appeared to be the only fraction capable of reducing the allergen-induced β-hex release (*p* = 0.0084; Figure 6C). Raw milk fraction 3 showed a tendency towards a reduction (*p* = 0.0794; Figure 6C).

## 4. Discussion

Previously, we observed that the suppression of local type 2 cytokine levels, produced by other immune cells than T cells, seemed to be crucial for the raw milk-induced prevention of allergic asthma in a murine house dust mite-induced asthma model [11]. Since it is well known that type 2 cytokines can also be produced by mast cells and basophils [24,25], this hinted towards a potential role of these effector cells in the allergy-protective effect. The present study therefore extensively investigated the direct effects of raw cow’s milk on murine mast cells in vitro.

In this study, we demonstrate that raw cow’s milk has the capacity to inhibit the allergic effector response in vitro, by directly affecting mast cell activation. Exposing BMMC and PMC to raw milk prior to activation significantly reduced β-hex release and de novo IL-6 and IL-13 production. Since ionomycin-induced mast cell activation was not affected by raw milk, raw milk potentially inhibits the activation of the proximal signaling pathway of the FcεRI. Raw milk-treated mast cells furthermore retained membrane-bound IgE expression after allergen stimulation. This may indicate that raw milk exerts its effects by reducing the translocation of engaged FcεRI complexes into lipid rafts and subsequent receptor internalization, required for FcεRI-mediated mast cell responses. In line with previous epidemiological as well as preclinical studies showing a loss of allergy protection upon milk processing [6,11,12,13,17], heated raw milk and shop milk (store-bought milk) did not inhibit mast cell activation. Moreover, this study provides some insight into the raw milk components responsible for the reduced mast cell activation, by demonstrating that these components are likely to have a molecular weight of > 37 kDa.

As we wanted to build on previous findings observed in a murine allergic asthma model [11], murine mast cells were used. In mice, mucosal-type mast cells and connective tissue-type mast cells represent the two main mast cell phenotypes [20]. BMMC and PMC are respectively considered as their in vitro equivalents. IgE-mediated BMMC activation was significantly inhibited by raw cow’s milk, as illustrated by a reduced β-hex release and reduced de novo IL-6 and IL-13 production. In PMC, a similar reduction in β-hex release was observed after IgE-mediated stimulation. However, IgE-mediated stimulation of PMC did not result in the secretion of newly formed cytokines (data not shown). This is in line with previous studies demonstrating that PMC secrete no or small amounts of newly formed proinflammatory mediators upon activation [18]. The reduced BMMC activation coincided with reduced downstream signaling events of the FcεRI, such as SYK phosphorylation, induction of calcium influx and activation of ERK. Intracellular calcium influx, key in the activation pathway leading to mast cell degranulation, was significantly reduced by raw milk. SYK and ERK phosphorylation were lower in raw milk-treated BMMC compared to control BMMC, although differences did not reach full significance. SYK is activated momentarily after FcεRI crosslinking and its activation is crucial to relay further cellular activation. The relatively weak effect on SYK and ERK phosphorylation compared to the observed inhibition of degranulation and cytokine production may be due to possible additional effects of raw milk components on—for example—gene transcription and translation, but potentially also on posttranslational processes and eventual cytokine secretion, which were not assessed in this study.

Interestingly, ionomycin-induced BMMC activation was not affected by raw milk. Since ionomycin artificially increases intracellular calcium levels and thereby bypasses the proximal signaling pathway of the FcεRI, this suggests that raw milk probably acts on these earlier signaling events. Upon cross-linking of the FcεRI, the immunoreceptor tyrosine-based activation motifs (ITAMs) in the cytoplasmic domain of the β and γ receptor subunits are phosphorylated in a Lyn-dependent manner. The protein kinase Syk is then recruited to the phosphorylated ITAMs, where it becomes activated, and subsequently phosphorylates and activates other proteins in the signaling cascade [3]. Lipid rafts have been shown to play an important role in this early signaling process, as they function as platforms capable of assembling the signal transduction machinery [26,27,28]. Migration of the FcεRI to these lipid rafts upon allergen cross-linking has been demonstrated by several studies [22,23]. The fact that the FcεRI is soluble in Triton X-100 at a steady state, but not after cross-linking [29], provides an opportunity to investigate FcεRI migration to lipid rafts upon allergen-specific stimulation. By immunoblotting the Triton X-100 soluble fraction of cell lysates for IgE, we could demonstrate a reduction in the presence of membrane-bound IgE, with increasing DNP-HSA concentrations. This is in line with the previously mentioned studies [22,23], indicating an increased migration of the FcεRI to lipid rafts upon allergen-specific stimulation. Interestingly, compared to processed milk-treated BMMC, raw milk-treated BMMC tended to retain IgE expression on their cell membrane after activation, regardless of DNP-HSA concentration. Retaining membrane-bound IgE expression, and thus preventing FcεRI migration to lipid rafts, might therefore be a way by which raw cow’s milk was able to reduce IgE-mediated mast cell activation.

To gain more insight into the raw milk components responsible for the observed reduction in mast cell activation, raw milk was fractionated based on molecular size. Since the inhibition of mast cell activation was not observed with heated raw milk, indicating a loss of protection upon heat treatment, focus was on the heat-sensitive proteins. From the 24 raw milk fractions obtained, only fraction 2 was able to reduce β-hex release in the mast cell activation assay. Fraction 3 showed a tendency towards a reduction. According to the SDS-PAGE, these fractions contained mostly proteins with a molecular weight between 50 and 75 kDa. Interestingly, the most abundant milk proteins, i.e., the caseins, β-lactoglobulin, and α-lactalbumin, all have a molecular weight of < 25 kDa. This indicates that these proteins are probably not responsible for the observed reduction in mast cell activation. In addition, this suggests that the observed effects are likely to be attributable to less abundant proteins with immunomodulatory functionalities. Lactoferrin (80 kDa), monocyte differentiation antigen CD14 (56 kDa), polymeric immunoglobulin receptor (83 kDa), and lactadherin (~50 kDa) are some examples of proteins often mentioned in relation to the allergy-protective effect of raw cow’s milk [15,30], that fall within the molecular weight range. In addition, fraction 2 and 3 also show a clear protein band around 150 kDa, indicating that IgG (150 kDa) and alkaline phosphatase (140 kDa) could also be involved. Alkaline phosphatase is particularly interesting, as we have previously demonstrated that it has the capacity to restore the allergy-protective effects abolished by heat treatment in a murine ovalbumin-induced food allergy model [12]. However, further research on the characterization of the protective fractions by proteomic analysis is needed to gain more insight into the actual raw milk protein(s) involved.

In the in vivo situation, direct contact between mast cells and raw milk is unlikely. During passage through the gastrointestinal tract, raw milk will be, at least partly, degraded. Moreover, mast cells in the gut are located beneath the epithelial surface [31], which hinders direct interaction between raw milk and mast cells. However, for several raw milk components, such as lactoferrin and TGF-β, it has been demonstrated that they can survive passage through the gastrointestinal tract upon ingestion [14]. Whether these components can directly (e.g., due to epithelial barrier disruption as demonstrated in children with food allergy [32] or via transepithelial uptake) or indirectly (e.g. via modulation of epithelial cells or the gut microbiome) affect mast cells should be assessed in future studies.

In conclusion, we demonstrate a direct inhibition of the allergic effector response by raw cow’s milk. Next, to the already described capacity to modulate T cell responses, the present study shows that raw cow’s milk is also able to directly influence mast cell activation. Mast cell activation was not affected by heated raw milk and shop milk, supporting the current evidence for a loss of allergy protection after milk processing and more specifically, after heat treatment. By fractionating raw milk based on molecular size, we could demonstrate that the heat-sensitive raw milk components responsible for the reduced mast cell activation are likely to have a molecular weight of > 37 kDa. In addition, we showed that raw milk potentially affects mast cell activation by acting on the proximal signaling pathway of the FcεRI. Raw milk-treated mast cells tended to retain membrane-bound IgE expression after allergen stimulation, compared to processed milk-treated mast cells, suggesting that raw milk reduced FcεRI migration to lipid rafts, crucial for FcεRI-mediated signal transduction. The evidence for a direct inhibition of mast cell activation by raw cow’s milk, provided in this study, extends the current knowledge on mechanisms underlying the allergy-protective effects of raw cow’s milk. Since raw cow’s milk consumption is discouraged because of the possible contamination with pathogens, a better understanding of these mechanisms is key for the development of microbiologically safe alternatives.

## Figures and Tables

**Figure 1 cells-09-01258-f001:**
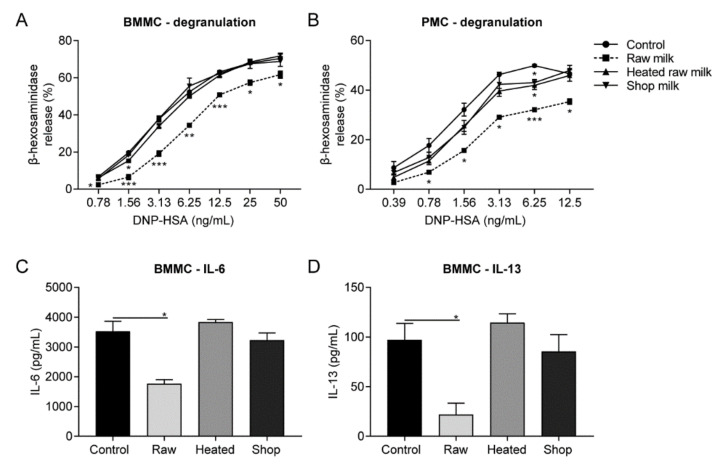
IgE-mediated mast cell activation reduced by raw milk. Primary mouse mast cells were incubated overnight with raw milk, heated raw milk, or shop milk, before they were primed with DNP-specific IgE and stimulated by a range of DNP conjugated to human serum albumin (DNP-HSA) concentrations. β-hexosaminidase release by (**A**) bone marrow-derived mast cells (BMMC) and (**B**) peritoneal mast cells (PMC) measured in supernatant, collected 1 h after DNP-HSA stimulation. (**C**) IL-6 and (**D**) IL-13 production by BMMC measured in supernatant collected 18 h after DNP-HSA stimulation. Data are presented as mean ± SEM and are representative of three independent experiments. * *p* < 0.05, ** *p* < 0.01, *** *p* < 0.001, compared to the control group as analyzed with one-way ANOVA followed by Dunnett’s multiple comparisons test. BMMC, bone marrow-derived mast cells; PMC, peritoneal mast cells; DNP-HSA, 2,4-dinitrophenol conjugated to human serum albumin; raw, raw cow’s milk; heated, heated raw cow’s milk; shop, shop milk.

**Figure 2 cells-09-01258-f002:**
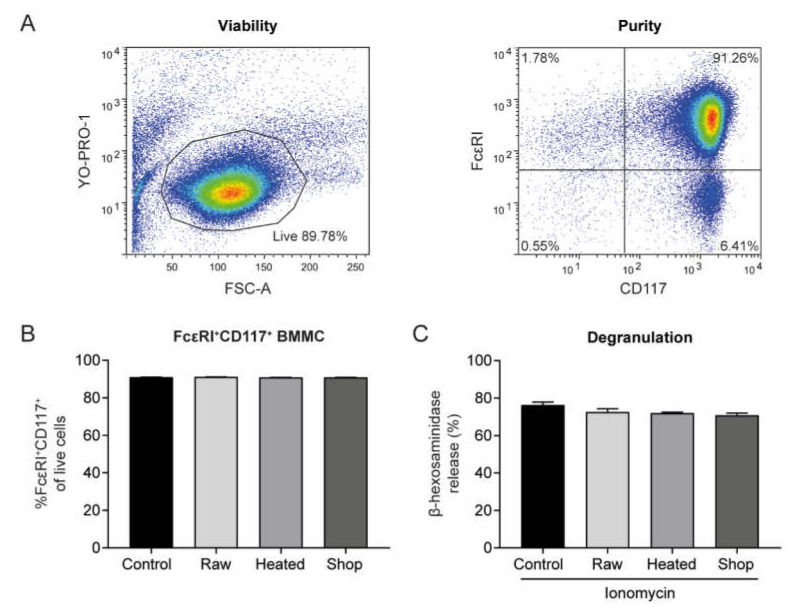
Raw milk exposure did not affect cell viability, FcεRI expression, and ionomycin-induced mast cell activation. (**A**) Viability and purity of cultured BMMC. (**B**) FcεRI and CD117 expression on BMMC after overnight milk exposure. (**C**) Ionomycin-induced BMMC activation after overnight milk exposure. Data are presented as mean ± SEM and are representative of three independent experiments. No significant differences were observed. FSC-A, forward scatter-area; BMMC, bone marrow-derived mast cells; raw, raw cow’s milk; heated, heated raw cow’s milk; shop, shop milk.

**Figure 3 cells-09-01258-f003:**
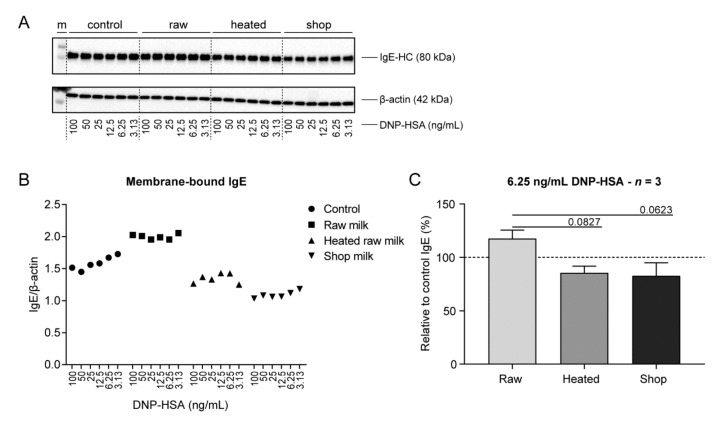
Membrane-bound IgE expression retained after DNP-HSA stimulation in raw milk-treated BMMC. After incubation with the different milk types and IgE-mediated activation by a range of DNP-HSA concentrations, BMMC were directly lysed in Triton X-100 for SDS-PAGE and immunoblotting. (**A**) Triton X-100 soluble fraction of BMMC lysates immunoblotted for IgE and β-actin. (**B**) Densitometric values of IgE normalized to β-actin. (**C**) Membrane-bound IgE expression relative to the control group at 6.25 ng/mL DNP-HSA. Densitometric values are representative of three independent experiments (**A**–**B**). Membrane-bound IgE expression at 6.25 ng/mL DNP-HSA is presented as mean ± SEM of three independent experiments (*n* = 3; **C**). Statistical analysis was performed compared to the raw milk group using one-way ANOVA, followed by Dunnett’s multiple comparisons test. M, marker; raw, raw cow’s milk; heated, heated raw cow’s milk; shop, shop milk; IgE-HC, IgE heavy chain; DNP-HSA, 2,4-dinitrophenol conjugated to human serum albumin.

**Figure 4 cells-09-01258-f004:**
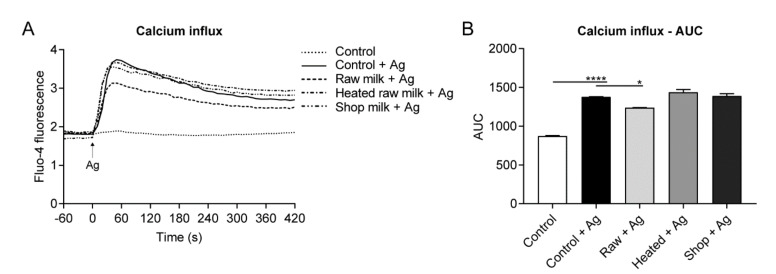
Raw milk-treated BMMC decreased calcium influx upon allergen challenge. (**A**) Changes in Fluo-4 fluorescence, representing changes in intracellular calcium levels, measured at baseline (for 60 s) and after allergen stimulation (indicated by the arrow; for 420 s). A representative graph is shown. (**B**) AUC analysis of the calcium influx, presented as mean ± SEM. Data are representative of three independent experiments. * *p* < 0.05, **** *p* < 0.0001, compared to the allergen-stimulated control group (control + Ag), as analyzed with one-way ANOVA, followed by Dunnett’s multiple comparisons test. Ag, allergen; control, unstimulated control BMMC; control + Ag, allergen-stimulated control BMMC; raw + Ag, allergen-stimulated raw milk-treated BMMC; heated + Ag, allergen-stimulated heated raw milk-treated BMMC; shop + Ag, allergen-stimulated shop milk-treated BMMC; AUC, area under the curve.

**Figure 5 cells-09-01258-f005:**
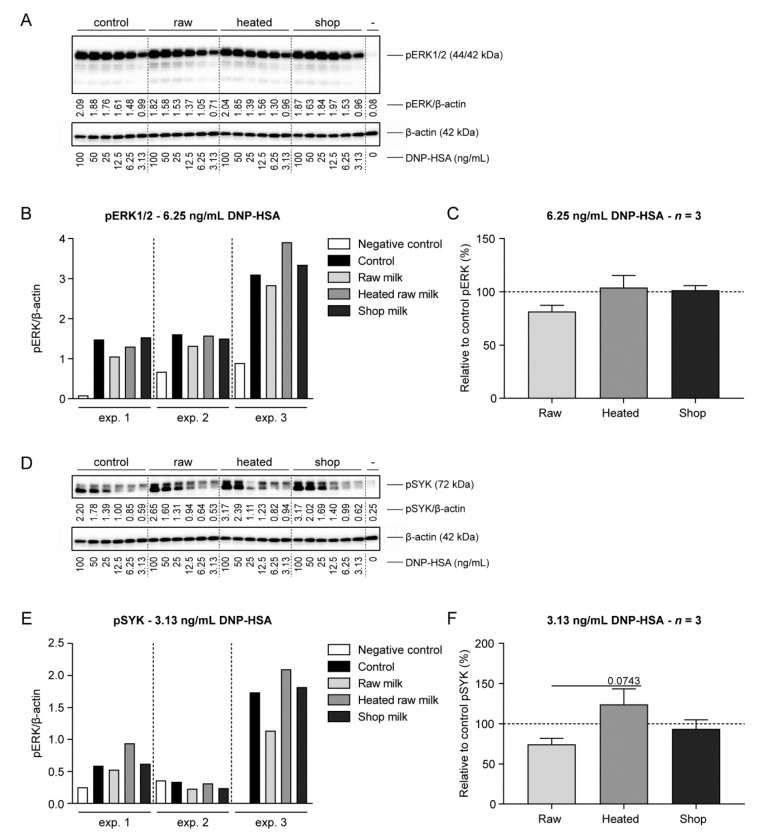
Lower ERK and SYK phosphorylation in BMMC treated with raw milk. After milk incubation and IgE priming, BMMC were incubated for 3 h without serum, to reduce autophosphorylation. Cells were subsequently stimulated for 10 min with DNP-HSA, after which phosphorylation was stopped and cells were lysed using SDS sample loading buffer. (**A**) Immunoblots for pERK and β-actin, including the densitometric values of the pERK/β-actin ratio at each DNP-HSA concentration (the immunoblots from experiment 1 are shown). (**B**) Densitometric values of the pERK/β-actin ratio at 6.25 ng/mL DNP-HSA, from three independent experiments. (**C**) ERK phosphorylation relative to the control group at 6.25 ng/mL DNP-HSA, presented as mean ± SEM of three independent experiments (*n* = 3). (**D**) Immunoblots for pSYK and β-actin, including the densitometric values of the pSYK/β-actin ratio at each DNP-HSA concentration (the immunoblots from experiment 1 are shown ). (**E**) Densitometric values of the pSYK/β-actin ratio at 3.13 ng/mL DNP-HSA, from three independent experiments. (**F**) SYK phosphorylation relative to the control group at 3.13 ng/mL DNP-HSA, presented as mean ± SEM of three independent experiments (*n* = 3). Statistical analysis was performed compared to the raw milk group using one-way ANOVA followed by Dunnett’s multiple comparisons test. Raw, raw cow’s milk; heated, heated raw cow’s milk; shop, shop milk; pERK, phosphorylated ERK; DNP-HSA, 2,4-dinitrophenol conjugated to human serum albumin; exp., experiment; pSYK, phosphorylated SYK.

**Figure 6 cells-09-01258-f006:**
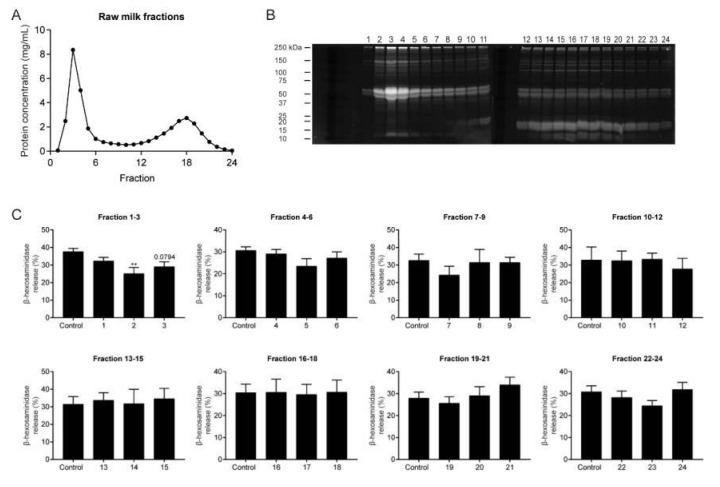
Reduction in IgE-mediated mast cell activation by raw milk fractions 2 and 3. (**A**) Chromatogram of proteins eluded from the gel-filtration column used to fractionate raw milk based on molecular size. (**B**) SYPRO^®^ Ruby Protein Gel Stain of raw milk fractions loaded onto an SDS-PAGE. (**C**) β-hexosaminidase release by BMMC incubated with raw milk fractions prior to IgE-mediated activation. Results of the protein chromatogram and SYPRO^®^ Ruby Protein Gel Stain are representative of three independent experiments. Results of the mast cell activation assay are presented as mean ± SEM of three independent experiments (*n* = 3). ** *p* < 0.01 compared to the control group as analyzed with one-way ANOVA, followed by Dunnett’s multiple comparisons test.

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
