# Peer review of "Direct Inhibition of the Allergic Effector Response by Raw Cow’s Milk—An Extensive In Vitro Assessment"

_cells, 2020, doi:10.3390/cells9051258_

Round 1
Reviewer 1 Report
It is a very interesting study that describes that raw milk is able to inhibit in vitro mast cell activation, and also authors propose some mechanisms of action involved. The manuscript is well written, clear and easy to read and methodologicaly adequate.
Some questions to be addressed by authors:
In Materials and Methods section:
- Line 81: please, indicate centrifugation speed conditions in "g" force, to methodology be reproducible by others.
In results section:
- In line 40, please indicate in the text the numeric data of BMMC viability (%), although these results are not graphicaly represented.
- In relation to assays about membrane-bound IgE expression after allergen-specific stimulation, statistic analysis is missing. Authors indicate that results are representative of al least three independent experiments, so statistic analysis must be done. I suggest to include a graph representing IgE/β actin OD obtained incubating BMMCs with control conditions, raw milk, heated raw milk and shop milk, at 12.5 ng/mL DNP-HSA stimuli. Data shoud be presented as mean±SEM and Dunnett’s multiple comparisons test done in relation to control condition. Modify discussion section if it is necessary (if there is no a significant effect)
- In relation to figure 5B, result should be presented as mean±SEM in both graphs and Dunnett’s multiple comparisons test done in relation to control condition. Modify discussion section if it is necessary (if there is no a significant effect)
- In Figure 6C, fraction 1-3. Statistic symbology is confusing. Please delete the upper line and locate ** on top of number 2 column.
In discussion section:
- line 66: what is the meaning of "As we wanted to elaborate on previous findings...... Please, clarify this sentence.
- Enrich the discussion section by suggesting which raw milk-described protein with MW between 55-70 kDa might be the one causing the inhibitory effect.
- In figure 6B, protein bands between 55-70 kDa are more intense (so more amount of protein is present) in fractions 3 and 4 than in fraction 2. Authors suggest that these proteins are causing the inhibitory effect. Discuss why there is no an inhibitory effect incubating cells with factions 3 and 4.
Author Response
The authors would like to thank reviewer 1 for the valuable comments on our manuscript. We address the questions raised by reviewer 1 by answering the specific comments point-by-point.
Reviewer: 1
It is a very interesting study that describes that raw milk is able to inhibit in vitro mast cell activation, and also authors propose some mechanisms of action involved. The manuscript is well written, clear and easy to read and methodologically adequate.
The authors would like to thank reviewer 1 for the valuable comments on our manuscript. We address the questions raised by reviewer 1 by answering the specific comments point-by-point.
Some questions to be addressed by authors:
In Materials and Methods section:
Line 81: please, indicate centrifugation speed conditions in "g" force, to methodology be reproducible by others.
As suggested, centrifugation speed is indicated in g force (as marked with track changes; page 2, line 81).
In results section:
In line 40, please indicate in the text the numeric data of BMMC viability (%), although these results are not graphically represented.
The graph representing BMMC viability (%) is added to the manuscript as supplementary figure (Figure S1B) (as marked with track changes; page 6, line 244 & page 15, line 477-488).
In relation to assays about membrane-bound IgE expression after allergen-specific stimulation, statistic analysis is missing. Authors indicate that results are representative of al least three independent experiments, so statistic analysis must be done. I suggest to include a graph representing IgE/β actin OD obtained incubating BMMCs with control conditions, raw milk, heated raw milk and shop milk, at 12.5 ng/mL DNP-HSA stimuli. Data shoud be presented as mean±SEM and Dunnett’s multiple comparisons test done in relation to control condition. Modify discussion section if it is necessary (if there is no a significant effect)
Since the IgE/β-actin OD/mm2 values differed quite a bit between the different experiments (due to the use of primary cells there can be quite some differences between batches), we initially decided to show 1 representative graph. However, we do agree with reviewer 1 that statistical analysis is very important. To be able to combine the results of the independent experiments, we therefore converted the results into percentages. In this way statistical analysis could be performed. This new graph is added to the manuscript (page 8, line 275) and the text is adapted accordingly (page 7, line 273-274 & page 8, line 281-288).
In relation to figure 5B, result should be presented as mean±SEM in both graphs and Dunnett’s multiple comparisons test done in relation to control condition. Modify discussion section if it is necessary (if there is no a significant effect)
For the same reasons as indicated above, we can only present a n = 3 graph by using percentages. However, we know that, depending on the batch, the BMMC activation pattern can differ. It can, for example, reach its optimum at a different DNP-HSA concentration. Accordingly, also the phosphorylation pattern differs and the effects we see do therefore not always happen as strong at the same concentration. Therefore, we added the results of the three experiments independently (page 10, line 327). The text is adapted accordingly (page 11, line 334-346) and the statements on the effects of ERK phosphorylation in the discussion section are toned down (page 13, line 397-408).
In Figure 6C, fraction 1-3. Statistic symbology is confusing. Please delete the upper line and locate ** on top of number 2 column.
The upper line is deleted, and the statistic symbology is located on top of the corresponding bars (page 12, line 359).
In discussion section:
line 66: what is the meaning of "As we wanted to elaborate on previous findings...... Please, clarify this sentence.
We meant to say that we wanted to build on previous findings, so to extend them. We clarified the sentence (page 13, line 389).
Enrich the discussion section by suggesting which raw milk-described protein with MW between 55-70 kDa might be the one causing the inhibitory effect.
As suggested by reviewer 1, we added some speculation about the raw milk protein(s) involved in the inhibitory effect to the discussion section (page 14, line 443-452).
In figure 6B, protein bands between 55-70 kDa are more intense (so more amount of protein is present) in fractions 3 and 4 than in fraction 2. Authors suggest that these proteins are causing the inhibitory effect. Discuss why there is no an inhibitory effect incubating cells with factions 3 and 4.
From the 24 raw milk fractions obtained, only fractions 2 and 3 were able to reduce β-hex release. Since these fractions mostly contained proteins with a molecular weight between 50 and 75 kDa we indeed suggested that proteins with this molecular weight are probably responsible for the inhibitory effect. However, this does not mean that these fractions contain the same proteins. Which proteins end up in which fraction is until now not known. Therefore, it could be very well possible that the inhibitory raw milk protein(s) involved is present in fraction 2, but absent in fraction 4. These are speculations and as mentioned proteomic analysis of these fractions will hopefully give more insight in the actual raw milk protein(s) involved.
Reviewer 2 Report
In this work, Abbring, S., et al. aim to analyze the possible allergy-protective effects of raw cow´s milk by directly measuring the effects of that aliment on FceRI-mediated mast cell activation in vitro. Authors find that incubation of murine bone marrow derived mast cells (BMMCs) and peritoneal-derived mast cells (PMCs) with raw cow´s milk for 24 hours diminished the IgE/Ag-induced beta hexosaminidase release, and IL-6 and IL-13 production. The inhibition seems to be related to the blockage of translocation of FceRI receptor to lipid rafts, diminished calcium movilization and lower ERK1/2 phosphorylation. After fractionation of protein compounds of raw cow´s milk, they find that a protein fraction (approximately of 37 kDa) seems to be responsible for the inhibition of IgE/Ag-induced beta hexosaminidase release.
Although the study is pertinent and results are of potential interest, there are some aspects of the experimental design and interpretation of the results that must be clarified.
- Authors should justify the times that were used to measure distinct parameters of cell activation. For example, why one hour of IgE-mediated cell sensitization was chosen? Why the 24 h time for incubation with raw cow´s milk was selected? Was 24 hours of incubation the time for maximal inhibition of IgE/Ag-induced beta hexosaminidase release? Why the time for detection of cytokines (IL-6 and IL-13) was 18 hours? What was the time of stimulation in the experiment shown in figure 3? Was this time the optimal to observe internalization of membrane bound IgE? Were the chosen times optimal for both (mucosal-type and connective tissue-type) mast cells?
- Authors should explain why the chosen concentration of milk was 5% vol/vol.
- The amount of IgE on the surface of control BMMCs after FceRI stimulation seems to decrease at higher concentrations of antigen (Figure 3B) and this is not observed in cells pre-treated with raw cow´s milk. However, no statistical analysis is shown. Does this indicate that differences on membrane-bound IgE (in control and treated cells) were not statistically significant? Also, in distinct parameters, heated raw milk and shop milk did not produce any effect, but seem to promote lower expression of membrane IgE, even at low amounts of antigen. In order to clarify this, statistical analysis (showing differences between antigen concentrations and between experimental groups) should be shown.
- Authors mention that the observed inhibition of IgE/Ag-induced beta hexosaminidase release and IL-6 and IL-13 production by raw cow´s milk incubation was significant (40%, 50% and 75% respectively). However, the effects of raw cow´s milk incubation on IgE/Ag-induced calcium rise were discrete and, on ERK1/2 phosphorylation, were not significant. How the authors explain those results if, as stated, degranulation and cytokine synthesis are related to calcium movilization and MAPK activation? Could another mechanism of raw cow´s milk inhibition be considered? This analysis must be included in the discussion.
- To date, several inhibitory compounds of mast cell activation (many of them from natural origin) have been described. The chemical structure of those compounds is diverse and many of them are not proteins (i.e. flavonoids, xantones, etc). For a better comprehension of the experimental design, authors should present in the introduction the reason why a protein compound was thought to be responsible for the inhibition produced by raw cow´s milk.
Author Response
The authors would like to thank reviewer 2 for the valuable comments on our manuscript. We address the questions raised by reviewer 2 by answering the specific comments point-by-point.
Reviewer: 2
In this work, Abbring, S., et al. aim to analyze the possible allergy-protective effects of raw cow´s milk by directly measuring the effects of that aliment on FceRI-mediated mast cell activation in vitro. Authors find that incubation of murine bone marrow derived mast cells (BMMCs) and peritoneal-derived mast cells (PMCs) with raw cow´s milk for 24 hours diminished the IgE/Ag-induced beta hexosaminidase release, and IL-6 and IL-13 production. The inhibition seems to be related to the blockage of translocation of FceRI receptor to lipid rafts, diminished calcium movilization and lower ERK1/2 phosphorylation. After fractionation of protein compounds of raw cow´s milk, they find that a protein fraction (approximately of 37 kDa) seems to be responsible for the inhibition of IgE/Ag-induced beta hexosaminidase release.
Although the study is pertinent and results are of potential interest, there are some aspects of the experimental design and interpretation of the results that must be clarified.
The authors would like to thank reviewer 2 for the valuable comments on our manuscript. We address the questions raised by reviewer 2 by answering the specific comments point-by-point.
Authors should justify the times that were used to measure distinct parameters of cell activation. For example, why one hour of IgE-mediated cell sensitization was chosen? Why the 24 h time for incubation with raw cow´s milk was selected? Was 24 hours of incubation the time for maximal inhibition of IgE/Ag-induced beta hexosaminidase release? Why the time for detection of cytokines (IL-6 and IL-13) was 18 hours? What was the time of stimulation in the experiment shown in figure 3? Was this time the optimal to observe internalization of membrane bound IgE? Were the chosen times optimal for both (mucosal-type and connective tissue-type) mast cells?
In our lab, we have a lot of experience with primary mast cell culture (both BMMC and PMC) and with the mast cell activation assays conducted. When this assay was developed, we determined and optimized the incubation times for IgE priming (1 h) and DNP-HSA incubation (1 h) as well as the time for the detection of cytokines (± 18 h). The used incubation times are the ones of the standard protocol and will generally result in a mast cell degranulation of about 60%.
Mast cells were incubated overnight with raw cow’s milk. We started the experiments with 2 h raw milk incubation, after which we observed a small inhibition in the β-hexosaminidase release. To enhance the effect, we decided to incubate the mast cells overnight with raw milk, which resulted in a reduction in β-hexosaminidase release of about 40% (as illustrated in the manuscript). We never tried to incubate the cells for a longer period of time to see whether the effects could be even stronger.
For the experiment shown in Figure 3, the standard mast cell activation protocol was used (1 h IgE priming and 1 h DNP-HSA activation, as mentioned in the materials & methods section page 4, line 165-168). With this protocol we achieve a mast cell degranulation of about 60%. Changing these incubation times will not result in a higher mast cell degranulation. Therefore, we believe that these incubation times are also optimal for observing internalization of membrane-bound IgE.
- Authors should explain why the chosen concentration of milk was 5% vol/vol.
Just as for the incubation time, we optimized the raw milk concentration used. We tested raw milk concentrations between 1.25-20% v/v and after taking into account the effects observed and the viability of the cells, we decided to continue with 5% v/v. - The amount of IgE on the surface of control BMMCs after FceRI stimulation seems to decrease at higher concentrations of antigen (Figure 3B) and this is not observed in cells pre-treated with raw cow´s milk. However, no statistical analysis is shown. Does this indicate that differences on membrane-bound IgE (in control and treated cells) were not statistically significant? Also, in distinct parameters, heated raw milk and shop milk did not produce any effect, but seem to promote lower expression of membrane IgE, even at low amounts of antigen. In order to clarify this, statistical analysis (showing differences between antigen concentrations and between experimental groups) should be shown.
As primary cells behave a little different every batch you culture it was not possible to take an average of the three experiments conducted without introducing a lot of variation due to differences in OD/mm2 values between experiments. Therefore, we originally decided to show 1 representative graph, with the limitation of not being able to perform statistics. Since we agree with reviewer 2 that statistical analysis is very important, we added a figure to the manuscript showing the percentage of membrane-bound IgE (as average of three independent experiments) relative to the control group at 6.25 ng/mL DNP-HSA (page 8, line 275). - Authors mention that the observed inhibition of IgE/Ag-induced beta hexosaminidase release and IL-6 and IL-13 production by raw cow´s milk incubation was significant (40%, 50% and 75% respectively). However, the effects of raw cow´s milk incubation on IgE/Ag-induced calcium rise were discrete and, on ERK1/2 phosphorylation, were not significant. How the authors explain those results if, as stated, degranulation and cytokine synthesis are related to calcium movilization and MAPK activation? Could another mechanism of raw cow´s milk inhibition be considered? This analysis must be included in the discussion.
We agree with reviewer 2 that effects on ERK1/2 phosphorylation are not very big. However, for the calcium influx we should consider that statistical analysis was performed on the AUC. This AUC includes the first critical period of Ca2+ flux upon allergen challenge (60 seconds of baseline measurements and 420 seconds of measurements after allergen stimulation).
Regarding ERK1/2 phosphorylation, cells were incubated for 3 h without serum to reduce autophosphorylation. This could have resulted in a less strong effect. Nevertheless, we agree with reviewer 2 that also other mechanisms could contribute to the observed reduction in cytokine production. ERK1/2 phosphorylation is related to cytokine synthesis but not the only factor underlying actual cytokine production. There are many steps in between ERK1/2 phosphorylation and cytokine production such as gene transcription and translation, but also posttranslational processes and eventual secretion. These are all not investigated in the current manuscript and could have contributed to the discrepancy between ERK1/2 phosphorylation and cytokine production. This line of reasoning is added to the discussion section of the manuscript (page 13, line 404-408).
- To date, several inhibitory compounds of mast cell activation (many of them from natural origin) have been described. The chemical structure of those compounds is diverse and many of them are not proteins (i.e. flavonoids, xantones, etc). For a better comprehension of the experimental design, authors should present in the introduction the reason why a protein compound was thought to be responsible for the inhibition produced by raw cow´s milk.
By using preclinical murine models, we previously demonstrated that raw cow’s milk can prevent the development of both asthma and food allergy. This protective effect was clearly destroyed by heat treatment. In line with epidemiological studies and our preclinical results, heated raw milk did also not inhibit in vitro mast cell activation (as also mentioned in the manuscript, page 13, line 383-386). This clearly hints towards the contribution of a heat-sensitive raw milk component, making proteins likely candidates. This information is added to the manuscript (page 14, line 434-436).
Reviewer 3 Report
In this study Abbring and co-authors present their observations regarding inhibition of IgE-mediated mast cell (MC) activation by raw milk in vitro. The primary strengths of the research are that it logically follows on recent studies on the topic, further supports epidemiological interpretations [in the literature] with actual experimentation, and the authors use primary mast cell cultures. The experiments in this paper are generally well-executed, but there is some room for additional negative controls with immunoblotting and more precise characterization of raw milk proteins. The prose of the manuscript is very well written and easy to follow. I read the paper with great interest, and I look forward to its eventual publication; the following criticisms are offered in that spirit and I hope that the authors can reasonably address all of these points.
------------------------------------
- The authors do some preliminary work on mechanism, but this is restricted to comparison with ionomycin stimulus to show an effect more specific to IgE-FcεRI signaling. The pERK data are interesting but would be greatly enhanced by further analysis of the lysates for other constituents of FcεRI signaling, namely Fyn, Lyn, Syk, and Btk. Such observations would enrich mechanistic insights, especially as it might relate to downstream comparisons to other, more specific pharmacologic interventions (e., for investigators outside allergy). Additionally, these data are muted by not including the necessary non-activated (+IgE, but no DNP) controls – given that ERK is such a pleiotropic kinase, I am concerned that the effects seen could be due to mitogenic crosstalk or inhibition thereof, which would be a valuable observation too. While viability of the cells was shown, their potential for expansion over time was not examined. Unfortunately, I think this is a major limitation of the current work, and the simplest path to address it might be for the authors to perform additional immunoblots, hoping that there are leftover lysates.
- A major outstanding question – probably the most important – is: what is in raw milk fractions 2 and 3? REFs 6, 11-14 (same authors), and 17 discuss the protective effects of raw milk (omega-3 FAs for 17), and that the protection is lost with heating (but not skimming, REF 12) implying a likely protein component (REF 14). I am also pleased to see the authors consider gut transit and anatomical access in their discussion, where they have indicated some specific factors. Lactoferrin (80kDa) and TGF-β1 (44 kDa, latent) may or may not be abundant in fractions 2 & 3, further it is notable that active monomeric TGF-β1 is much smaller at ~12kDa. Altogether these points emphasize the need to deeply follow-up with more precise characterization of what is in the milk, and I strongly encourage the authors to include such data if they already have them in hand.
- Please explain/justify the sole use of female mice as the MC source.
- Please contextualize the biological relevance of 5% v/v milk in culture.
- I encourage any “data not shown” to be included as supplementary files.
- Data in Figure 3 could be supplemented by the FcεRI mean fluorescence intensities from Figure 2. Separate targets, but obviously closely related.
The following are not required experiments for the present work to be published, but I believe would be thoughtfully considered/discussed limitations for a more holistic work.
- Would one expect BMMC from C3H/HeOuJ mice to respond differently than those from Balb/c? There have been some papers in the past few years regarding whole animal allergy models using C3H/HeOuJ, they have mostly focused on the T cell-mediated response. If I recall correctly a hyporesponsive MC phenotype was described for non-IgE stimulus of MC from these animals (https://www.jimmunol.org/content/171/1/390). I don’t know if this was carefully examined for IgE-XL in this strain, though there was the knockout work reported in 129-C57 chimeras (https://www.jbc.org/content/287/11/8135.full).
- Have the authors tested their raw milk against cultures raised in IL-3 only? It has been reported that chronic exposure of mouse BMMC cultures to SCF attenuates Ag-IgE responses (https://www.jimmunol.org/content/188/11/5428.long). My lab also experiences this attenuation – conversely acute exposure promotes IgE-mediated response.
- In some locales unregulated raw milk shares are a serious source of antibiotic resistant food poisoning outbreak (for example, see 2018 jejuni case in southern Colorado; https://www.cdc.gov/mmwr/volumes/67/wr/mm6705a2.htm). I believe the authors and publisher need to be judicious in qualifying the results: I have seen too many layperson misinterpretations (and willful misrepresentations) of data from well-intentioned studies such as this. I think this risk is even greater with open access papers. I encourage consideration of precaution regarding the consumption of raw milk. In fact it is federally illegal to distribute raw milk across state lines in the USA, and outright sale is illegal within 21 states. This also underscores a need to be more precise about what is in raw milk conferring protective effect.
- Related to above, while I understand that it is extremely difficult given the types of cultures, a limitation of this work is that it is performed under antibiotic pressure, thus does not consider potential microbiotal protection.
Author Response
The authors would like to thank reviewer 3 for the valuable comments on our manuscript. We address the questions raised by reviewer 3 by answering the specific comments point-by-point, please see the attachment.

Round 2
Reviewer 1 Report
All the comments I made have been answered by authors, but I have some new ones about the revised version
In view of the results of the statistic analysis in relation to the effect of raw milk on membrane-bound IgE expression and ERK/SYK activation in BMMC, the manuscript clearly and strongly describe the inhibitory effect of raw milk in BMMC activation, but the mechanism is reduced to a decrease in calcium influx. Besides, there is no a (signifcant) effect of fraction 3 on mast cell activation. So sentences 24-25, 357, 380-381, 428-430, 437, 446, 470 are not properly explaining the results. Please, re-write them in accordance to results .
Author Response
As requested by reviewer 1, we rewrote the results and toned down the conclusions (page 1, line 23-27; page 11, line 359-362; page 13, line 382-384; page 14, 431-434; page 14, line 440-442; page 14, line 472-477).
Reviewer 2 Report
Authors have addressed all the concerns. There are no further comments.
Author Response
The authors would like to thank reviewer 2 for taking the time to thoroughly review our manuscript.
Reviewer 3 Report
The authors have satisfied my previous critical points. I look forward to the eventual publication of this work.
Author Response
The authors would like to thank reviewer 3 for taking the time to thoroughly review our manuscript.